# Investigation on the Evolution of Nano-Scale Defects of CL-20 Crystals under Thermal Treatment by Wide/Small-Angle X-ray Scattering

**DOI:** 10.3390/ma15124258

**Published:** 2022-06-15

**Authors:** Haobin Zhang, Hongfan Wang, Jinjiang Xu, Jie Sun, Xiaolin Wang

**Affiliations:** 1Institute of Nuclear Physics and Chemistry, China Academy of Engineering Physics, Mianyang 621900, China; zhhb03@caep.cn; 2Institute of Chemical Materials, China Academy of Engineering Physics, Mianyang 621900, China; whf00300@163.com (H.W.); xujinjiang@caep.cn (J.X.); sunjie@caep.cn (J.S.); 3China Academy of Engineering Physics, Mianyang 621900, China

**Keywords:** CL-20, crystal defects, WAXS, SAXS, specific surface, pore size distribution

## Abstract

Nano-scale crystal defects extremely affect the security and reliability of the explosive charges of weapons. In order to understand the evolution of nano-scale defects of 2,4,6,8,10,12-hexanitro-2,4,6,8,10,12-hexaaza-isowurtzitane (CL-20) explosive crystals under thermal treatments, the specific surface, volume fraction and size distribution of the nano-scale defects were studied by using Wide Angle X-ray Scattering (WAXS) and Small Angle X-ray Scattering (SAXS) during the temperature range from 30 °C to 200 °C. The results showed that the number and size of the pores in CL-20 powder did not change significantly during the heating process before phase transformation (30–160 °C). At 170 °C, CL-20 began to convert from ε- to γ- phase, and the specific surface and volume fraction of the nano-scale defects increased significantly. Further investigation of the pore size distribution showed that the number of pores with a small size (radius 9–21 nm) changed particularly significantly, resulting from the cracking of the CL-20 crystal powder during phase transition. At 200 °C, the phase transition was completed and γ-CL-20 was created, and the small-sized pores gradually grew into medium-sized (radius 21–52 nm) pores over time when the temperature was fixed at 200 °C.

## 1. Introduction

Under the background of the rapid development of modern military technology, the security and effectiveness of weapon systems in harsh environments have attracted wide attention. As the energy carrier of weapon systems, explosives play a key role in weapon damage ability. However, the initial defects such as vacancies, impurities and pores often occur in the preparation and growth of explosive crystals. Moreover, in the process of storage and transportation, external stimuli such as temperature and pressure also lead to a large number of pores in the explosive crystals, which seriously affect the safety and effectiveness of weapon systems. For polycrystalline explosives, the phase transition process also brings defects. The existence of these defects not only lead to the increase in potential hot spots [1,2,3] and shock wave sensitivity to explosives [4], but also the decrease in mechanical properties [5,6,7] and thermal stability [8,9], which are not conducive to the effective performance of a weapons system. To sum up, the generation and formation of defects in explosive crystals are widespread but important. In order to ensure the safety and reliability of weapon systems, it is urgent that the structure of explosive crystal defects is studied.

In recent years, many techniques have been used to characterize the defects of explosive crystals, such as Computed Tomography (CT) [10], Optical Microscopes (OM) [11], Scanning Electron Microscopy (SEM) [12] and plane-polarized light microscopy [13]. However, the nano-scale defects of explosive crystals are difficult to detect effectively because the defects are distributed in the explosive crystal with a large number but small scale, and the explosive material easily decomposes under the high-energy stimulation from characterization methods. To characterize the nano-scale defects, small angle X-ray/Neutron Scattering technology (SAXS/SANS) is used as an advanced means to obtain specific information of the explosive internal defects [14,15,16,17,18,19,20]. Willey [21] investigated the defects in PBX9501 and LX10 under thermal stimulation and found that both the PBX9501 and LX10 took place during phase transition and produced numerous pores, by using the ultra-small angle X ray scattering technique (USAXS). Yan [22,23,24] studied the change of pore size and quantity before and after the phase transition of HMX crystals and HMX-based PBX explosives by using the SAXS technology, and suggested that the phase transition is the main reason for the increase in defects. To sum up, in order to optimize the safety performance of explosive crystals, it is very important to study the change of defect structure during phase transition.

2,4,6,8,10,12-hexanitro-2,4,6,8,10,12-hexaaza-isowurtzitane (CL-20) is considered to be one of the best single explosives, with excellent comprehensive properties. It is the prominent representative of the third-generation explosives and the most promising high energy density compound, which is expected to replace the commonly used octahydro-1, 3, 5, 7-tetranitro-1, 3, 5, 7-tetrazocine (HMX) at present [25]. Nevertheless, CL-20 is a poly-crystalline explosive and easily converts from ε- to γ- phase during the temperature range 140–200 °C, resulting in the decrease in CL-20 density and the increase in volume, with the production of new pore groups [26,27,28]. Until now, there have been few studies on the pore structure changes during the phase transition of CL-20.

In the current work, we infiltrated CL-20 crystal powder in a perfluoropolyether (GPL-107) matching solution [29] and investigated the evolution of the nano-scale defects of CL-20 under non-isothermal phase changing and high temperature treatment by using in situ WAXS and SAXS techniques. GPL-107 perfluoropolyether reagent was chosen as the matching solution because of its chemical inertness and similar electron density pair to CL-20 crystals, which enabled us to use SAXS to study the pore structure without the influence of the crystal surface. Our study provides more information for the further understanding of the defect evolution of CL-20 under thermal stimulation.

## 2. Materials and Methods

### 2.1. Materials and Instruments

CL-20 crystal powder with a particle size of 30~60 μm in diameter was provided by the Institute of Chemical Materials, China Academy of Engineering Physics (CAEP). GPL-107 perfluoropolyether was purchased from Bonway Technology Co., Ltd. (Hong Kong). The electron density of ε-CL-20 crystal form is 622.7 nm^−3^ and that of γ-CL-20 is 584.8 nm^−3^, while the electron density of GPL107 is 571.7 nm^−3^.

The in situ WAXS and SAXS experiments were carried out on a Xeuss 2.0 system from Xencos France equipped with a multi-layer focused Cu Kα X-ray source (GenilX3D Cu ULD, Xencos SA, Paris, France) operating at a maximum power of 30 W. The wavelength of the X-ray radiation was 0.154 nm. Two pairs of scatterless slits were located 1200 mm apart from each other for collimating the X-ray beam. Scattering data were recorded with the aid of a Pilatus 300 K detector (DECTRIS, Swiss, resolution: 487 × 640, pixel size = 172 µm). A Linkam TST 250 stretch heating sample table was used to control the temperature of the in situ experiments. Background scattering and X-ray absorption correction were performed on all the SAXS data. The WAXS and SAXS spectra were analyzed and inverted to one-dimensional curves using Fit-2D software. McSAS software was used to calculate the aperture size distribution. 

### 2.2. Experimental Process

The CL-20 samples were filled evenly in a copper ring with a thickness of 1 mm and inner diameter of 2 mm, and the samples were fully infiltrated with a perfluoropolyether matching solution and then fixed with polyimide tape.

WAXS and SAXS testing was carried out synchronously during the temperature range from 30 to 200 °C, with a heating rate of 5 °C/min and one test every 10 °C. The testing time for each pattern was 1800 s. The sample-to-detector distance of WAXS and SAXS were 284 mm and 2486 mm, respectively. 

## 3. Results and Discussion

### 3.1. The Phase Transition of CL-20 Crystals by WAXS

WAXS is a useful means to characterize different crystal forms, and the WAXS patterns of the CL-20 crystals were collected at different temperatures, as shown in Figure 1. The WAXS two-dimensional patterns showed no changes between 30 °C and 160 °C. The changes appeared at 170 °C, with the appearance of new rings and the weakening of old rings, and the change ended at 200 °C. Notably, the rings at 30 °C were not stable nor successive because of the irregular orientation and large size of the CL-20 powder. As the temperature increases, the ring becomes more successive due to the random orientation of more CL-20 powder.

In order to characterize the change process quantitatively, the two-dimensional WAXS patterns of CL-20 were converted into one-dimensional curves, as shown in Figure 2. The WAXS curves of CL-20 at 160 °C are basically consistent with the original sample, which belong to ε-CL-20 crystal forms, indicating that no phase transition occurred before 160 °C. The diffraction peak of the γ-CL-20 phase was observed at 170 °C, and the diffraction peak of the ε-CL-20 phase was weakened simultaneously, indicating that the ε-γ phase transition occurred in the sample. Diffraction peaks of the ε-CL-20 phase did not completely disappear until 200 °C, indicating that the phase transition was completed at 200 °C.

### 3.2. The Evolution of Nano-Scale Defects of CL-20 Crystals at Different Temperatures

SAXS was used to characterize the nano-scale defects of the CL-20 crystals under different temperatures, as shown in Figure 3. It can be seen that the scattering intensity of CL-20 increases significantly from the SAXS patterns as the temperature increases and also over time when the temperature was fixed at 200 °C. This indicated that the number of nano-scale defects obviously increased during the thermal treatment. From the characterization of WAXS, the CL-20 took place ε-γ phase transition above 170 °C, so the phase transition causes the increase in nano-scale defects. 

Using the Fit 2D software, the 2D SAXS patterns were converted into 1D curves, as shown in Figure 4. The scattering intensity of the samples tested in SAXS is relative scattering intensity, which is closely related to external conditions, such as experimental conditions, sample size, instrument parameters, exposure time and so on. To accurately characterize the evolution of the pore structure of the CL-20 powder during the phase transition, we corrected the absolute scattering intensity, which is the ratio of the scattering intensity to the incident intensity, which is only related to the properties of the sample itself, and has nothing to do with the external conditions [30].

Figure 4 shows the SAXS curves of CL-20 powder at 30, 160, 170, 180, 190, and 200 °C. The increase in scattering intensity indicates an increase in the numbers of internal nano-scale pores. The change of the scattering curve represents the change of internal microstructure, with the small-sized void represented by a high *q* region, and the large-sized pore represented by a low *q* region. Before 160 °C, the scattering intensity increased with the increase in temperature, but the shape did not change, and the number of pores in each size increased uniformly. The scattering intensity of the high *q* region suddenly increased at 200 °C, which is the end of phase transition, indicating the formation of more small-sized voids. The specific surface area and volume fraction of the pores were calculated according to the absolute scattering intensity [29], and the results are shown in Figure 5. Between the temperature range of 30 °C and 160 °C, the specific surface area and volume fraction of the pores only increased 0.5 times, which was mainly affected by the thermal expansion of the CL-20 crystals. After 170 °C, the specific surface area and volume fraction of the pores increased significantly, until the end of phase transition at 200 °C, from 0.134% to 0.256%, and the specific surface area increased from 2001 cm^−1^ to cm 9104 cm^−1^, which proves that most of the CL-20 pores come from the phase transition process.

For further investigation of the pore size distribution of CL-20 powder before and after phase transition, we used the spherical model [31] to fit the SAXS data, and the fitting results are shown in Figure 6. During the temperature from 30 °C to 160 °C, the number of pores increased, but the size distribution did not change. At 170 °C, the pore size distribution began to change, with the number of small pores with a radius of 9–21 nm increasing faster than the medium-sized pores with a radius of 21–52 nm, corresponding to the raised area of the SAXS curve in Figure 6. From 170 °C to 200 °C, the pore size distribution tends to be stable, with only quantitative changes. This means that numerous small pores were produced in the phase transition. At 200 °C, the medium-sized pores is also noticeably increased, which indicates that more small pores were converted to medium-sized pores.

### 3.3. The Evolution of Nano-Scale Defects of CL-20 Crystals over Time at 200 °C

To further understand the effect of thermal stimulation on pore evolution, we fixed the temperature at 200 °C, which is the end of phase transition, to observe the pore evolution under high temperature treatment. Figure 7 shows the SAXS curves over time at 200 °C. The scattering intensity of the γ-CL-20 increased noticeably in 118 min at 200 °C, especially the intensity in the *q* region around 0.2 nm^−1^, which indicated that a group of small pores with a radius of about 30 nm had formed. To 198 min at 200 °C, the scattering intensity in the low *q* region increased noticeably, indicating that the small-sized pores gradually grew and fused into larger size voids. The scattering intensity in the low *q* region continued to increase slightly until 278 min at 200 °C, indicating that the medium and large size voids were still being produced. The results show that the CL-20 crystal is still greatly damaged over time at 200 °C.

These damages are mainly due to the thermal decomposition of the γ-CL-20. The gas production from the thermal decomposition process firstly causes the increase in small-sized pores, and the small-sized pores gradually grow into larger-sized pores as the gas product increases and the internal pressure of the crystal increases gradually. The specific surface area and volume fraction of the pores were calculated [29], and the results are shown in Figure 8. Pore volume fraction increased from 0.25% to 7.44% and specific surface area increased from 9104 cm^−1^ 122,763 cm^−1^.

The pore size distribution of the CL-20 crystals over time at 200 °C were fitted, as shown in Figure 9. From 38 min to 118 min at 200 °C, the number of small pores increased significantly. After 118 min, small-sized pores began to grow and fuse into medium- and large-sized pores. Until 278 min, the number of small-sized pores was basically stable, and the medium-sized and large-sized pores were still in a slowly increasing. The variation of the void size distribution of CL-20 powder during storage is consistent with that of the SAXS curve.

## 4. Conclusions

The evolution of the nano-scale pore structure of CL-20 crystals during the thermal treatment was studied by using in situ WAXS and SAXS synchronously. The results are as follows:(1)Thermal expansion phase (30–160 °C): The number of pores and the specific surface area changed slightly, and the number of defects in each size increased slightly with the increase in temperature. The change of defects in the CL-20 crystals before the phase transition is mainly due to the effect of the thermal expansion of the CL-20 crystal lattice;(2)Phase transition stage (170–200 °C): The volume fraction and specific surface area of the pores of the CL-20 crystals increased significantly during the phase transition, resulting mainly in small- and medium-sized pores, but the change of large-sized pores was not obvious. From the beginning of heating to the end of the phase transition, most of the pores came from the CL-20 phase transition process, and the effect of thermal expansion effect was not significant.(3)Thermal decomposition stage (over time at 200 °C): The specific surface area and volume fraction of CL-20 pores changed over time when the temperature was fixed at 200 °C. Small-sized pores first appeared, and medium-sized and large-sized pores subsequently appeared, indicating that small-sized pores gradually grow into medium-sized and large-sized pores.

During the thermal treatment, the nano-scale pores increased noticeably, which seriously increases the sensitivity of CL-20 and creates a danger to explosive charges with CL-20. To improve the application performance of CL-20, we should try to avoid an increase in such defects, such as by storing it at a constant low temperature to avoid thermal expansion and any phase transition.

## Figures and Tables

**Figure 1 materials-15-04258-f001:**
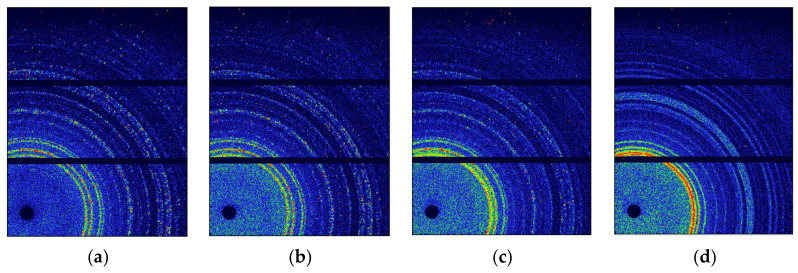
WAXS patterns of CL-20 heating at different temperatures: (**a**) 30 °C; (**b**) 160 °C; (**c**) 170 °C; (**d**) 200 °C.

**Figure 2 materials-15-04258-f002:**
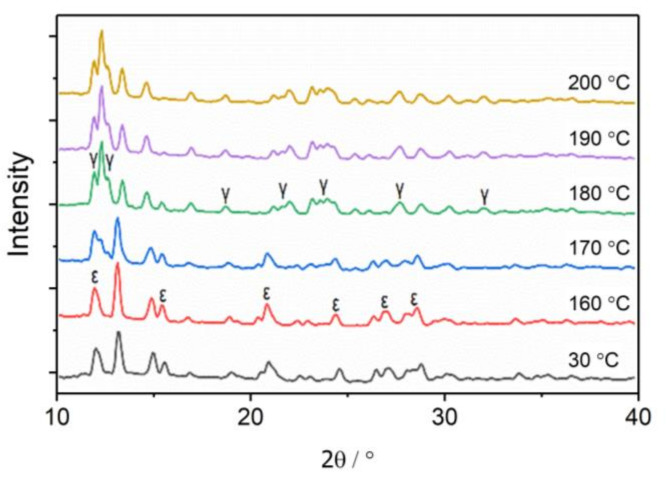
WAXS curves of CL-20 at different temperatures.

**Figure 3 materials-15-04258-f003:**
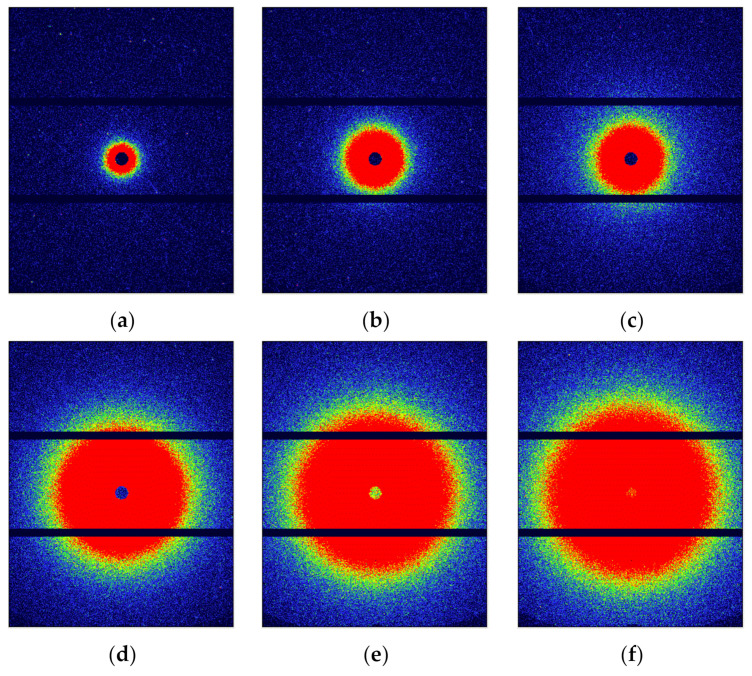
SAXS patterns of CL-20 after heating under different temperatures at different times: (**a**) 30 °C; (**b**) 170 °C; (**c**) 200 °C immediately; (**d**) 200 °C after 118 min; (**e**) 200 °C after 198 min; (**f**) 200 °C after 278 min. All the SAXS patterns are on the same strength scale.

**Figure 4 materials-15-04258-f004:**
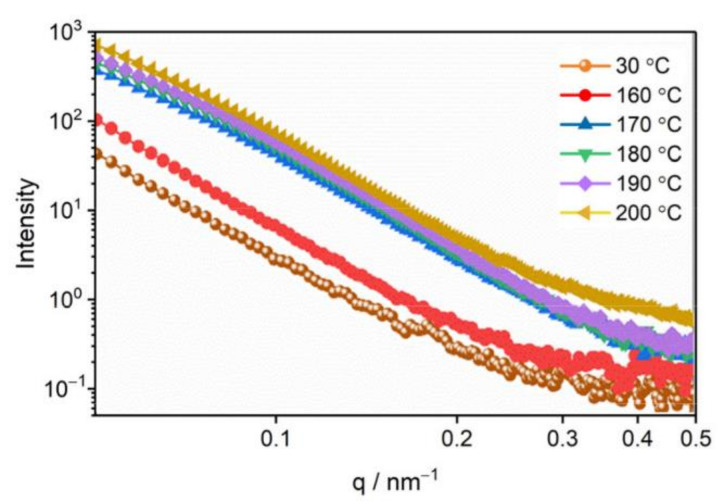
SAXS curves of CL-20 at different temperatures. In order to obtain the change trend of the scattering curve more clearly, the horizontal axis and the longitudinal axis are logarithmic.

**Figure 5 materials-15-04258-f005:**
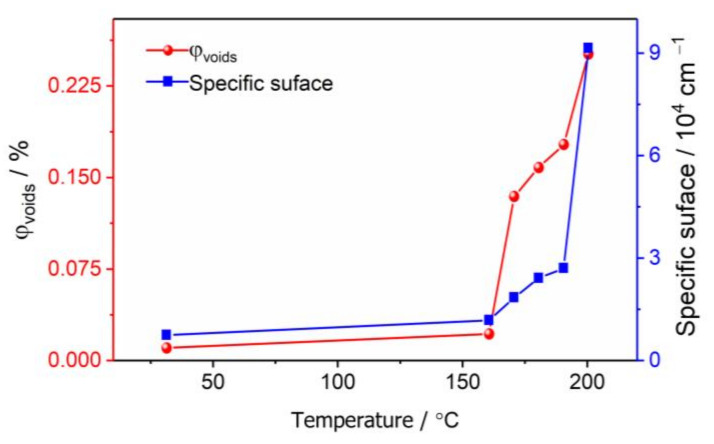
Pore volume fraction and specific surface of CL-20 before and during phase transition.

**Figure 6 materials-15-04258-f006:**
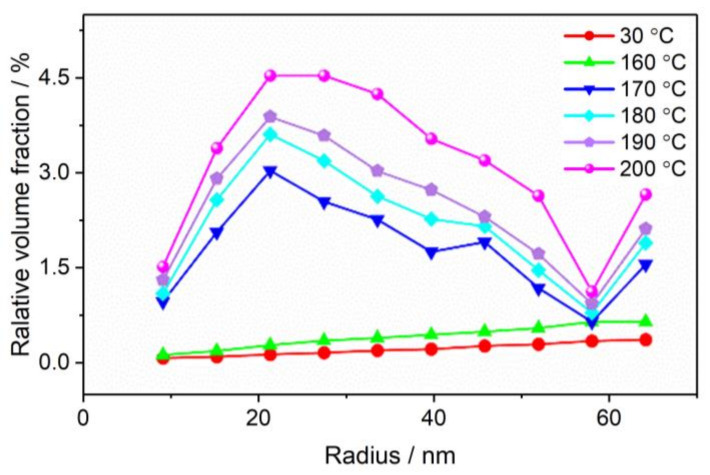
Pore size distribution of CL-20 before and during phase transition.

**Figure 7 materials-15-04258-f007:**
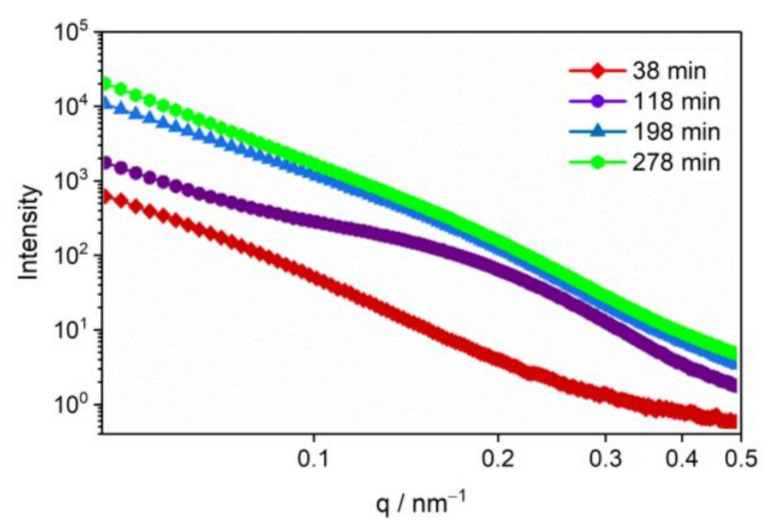
SAXS curves of CL-20 over time at 200 °C.

**Figure 8 materials-15-04258-f008:**
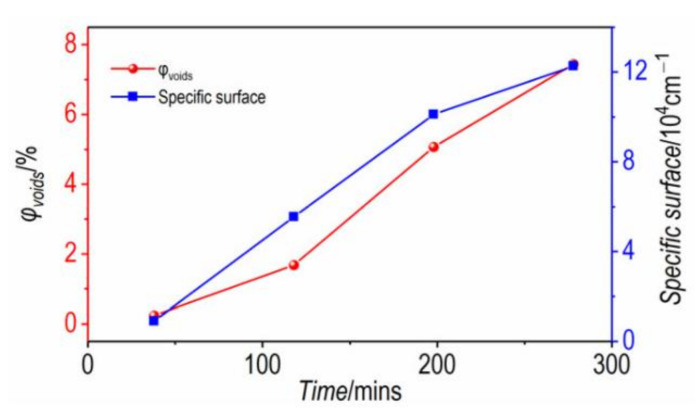
Pore volume fraction and specific surface of CL-20 over time at 200 °C.

**Figure 9 materials-15-04258-f009:**
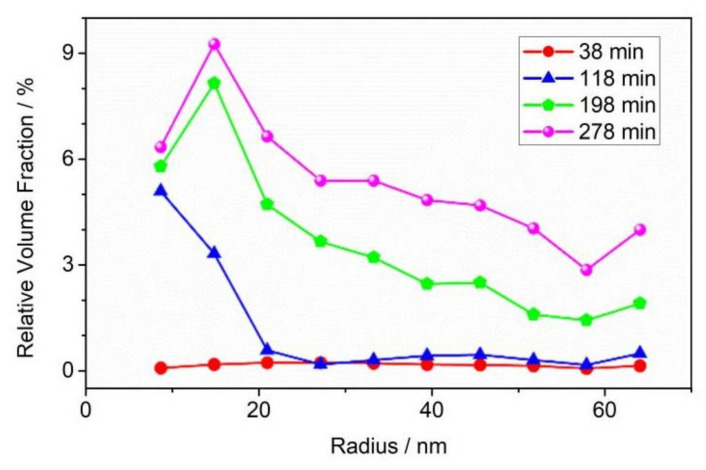
Pore size distribution of CL-20 as a function of storage time when the temperature is fixed at 200 °C.

## Data Availability

The data that support the findings of this study are available from the corresponding author upon reasonable request.

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
