# Peer review of "Investigation on the Evolution of Nano-Scale Defects of CL-20 Crystals under Thermal Treatment by Wide/Small-Angle X-ray Scattering"

_materials, 2022, doi:10.3390/ma15124258_

Round 1

Reviewer 1 Report

Review of article entitled: “Investigation on the Evolution of Nano-scale Defects of CL-20 2 Crystals under Thermal Treatment by Wide/Small-Angle X-ray 3 Scattering”

Overall, I liked the paper and believe it contributes to our knowledge of thermal evolution of CL-20. The authors used both WAXS and SAXS to examine CL-20 particles as they were heated from room temperature to around 200C. The analysis included determination of the number of voids and specific surface area during the heating process. I had several questions regarding this work:

1.     How do the WAXS and SAXS differ? How do these experiments differentiate between increasing the number of defects and the size of the defects?

2.     The authors mention cracks as well as voids. How are the cracks differentiated from the voids? Were voids observed? Could computed microtomography be used to confirm some of the experimental results?

3.     Are your pores assumed to be spherical? Are cracks cylindrical. Are they statistical in nature?

4.     I would be interesting to have additional support for the process by using thermal experiments such as differential scanning calorimetry (DSC). How do you results compare to phase transformation measured using DSC? Are the onsets and finishing of the phase transformations consistent with the DSC measurements?

5.     Do your experiments (WAXS and SAXS) measure the defect size and number of defects or is a model used to infer these quantities?

6.     What do the colors in your figures represent?

7.     Could there be any other explanation rather than increase in number of internal nano-scale pores? Could the increase be due to larger defects rather than an increase in the number of defects? If so, how do you prove this? Could you use X-ray microtomography on your samples?

8.     Is the temperature high enough to cause decomposition effects? If so, do the decomposition products form bubbles in your solvent? If so, could this bias your conclusions?

9.     Can you explain why an increase in scattering intensity indicates increase in more small voids rather than an increase in existing void sizes?

10.  Could you be more specific on the "spherical model" on line 170? Co you assume the pore volume remains unchanged, but the number increases? or visa verse?

11.  What do you mean by “are calculated” on line 199? What are the basic equations and assumptions for this calculation?

12.  Again, what is happening? Are the number of pores increasing or is the existing pores increasing in volume?

As you can tell by these questions, my main issue is related to the specific form of the thermal damage. Is it the number of defects that are increasing? Or, is it the volume of the defect that is increasing? Or, are both the number and size of the defect increasing. Do you have another proof for your conclusions. I also noticed that your references did not include any of the more recent work published in 2021 or 2022. There are some more recent  articles that would be worth citing.

Reviewer 2 Report

The authors present an experimental study of nano-scale defects in CL-20 under heating. Some changes in propagation of pores caused by both the phase transitions and temperature were observed by an appropriate experimental technique. Overall, the work is done well and manuscript sounds. Thus, I can recommend it for publication after a minor revision. In the introduction, the authors foreground their work, focusing on the relationship between the presence of pores and sensitivity. It indeed the case, but I could not find any discussion on this issue. Therefore, I suggest include some comments on this and, probably, provide some discussion of how to avoid the increase of such defects, which increase sensitivity of energetic crystals, in particular CL-20.

Reviewer 3 Report

The manuscript is devoted to research on the phase transformations of CL-20 at various temperatures. The nano-scale defects were studied by using Wide Angle X-ray Scattering (WAXS) and Small Angle X-ray Scattering (SAXS), during the temperature range from 30 °C to 200 °C .

 I recommend publishing the manuscript after minor the revision.

Comments on the manuscript.

 1.         Section 2.1. Materials and instruments, you need to specify the particle size of the initial CL-20 powders.

2.         “The electron density of ε-CL-20 crystal form is 622.7 nm-3 and that of γ-CL-20 is 584.8 nm-3, while the electron density of GPL107 is 571.7 nm-3” You need to specify the reference in which this data were obtained.

3.         Page 6. 199-198

“The specific surface area and volume fraction of pores are calculated, and the results are shown in figure 7.”  From the text of the manuscript it is not clear what method was used to obtain the calculated data on the specific surface area and volume fraction of pores.

4.         The conclusion must be supplemented with a forecast about the change in the performance of the SL-20 in terms of safety and suitability for use after temperature effects.
